# Valorization of Different Fractions from Butiá Pomace by Pyrolysis: H_2_ Generation and Use of the Biochars for CO_2_ Capture

**DOI:** 10.3390/molecules27217515

**Published:** 2022-11-03

**Authors:** Isaac dos S. Nunes, Carlos Schnorr, Daniele Perondi, Marcelo Godinho, Julia C. Diel, Lauren M. M. Machado, Fabíola B. Dalla Nora, Luis F. O. Silva, Guilherme L. Dotto

**Affiliations:** 1Research Group on Adsorptive and Catalytic Process Engineering (ENGEPAC), Federal University of Santa Maria, Roraima Avenue, 1000-7, Santa Maria 97105–900, Brazil; 2Department of Natural and Exact Sciences, Universidad de la Costa, CUC, Calle 58 # 55–66, Barranquilla 080002, Colombia; 3Postgraduate Program in Engineering Processes and Technology, University of Caxias do Sul—UCS, Caxias do Sul 95070-560, Brazil

**Keywords:** butiá wastes, pyrolysis, butiá biochar, H_2_ generation, CO_2_ adsorption

## Abstract

This work valorizes butiá pomace (*Butia capitata*) using pyrolysis to prepare CO_2_ adsorbents. Different fractions of the pomace, like fibers, endocarps, almonds, and deoiled almonds, were characterized and later pyrolyzed at 700 °C. Gas, bio-oil, and biochar fractions were collected and characterized. The results revealed that biochar, bio-oil, and gas yields depended on the type of pomace fraction (fibers, endocarps, almonds, and deoiled almonds). The higher biochar yield was obtained by endocarps (31.9%wt.). Furthermore, the gas fraction generated at 700 °C presented an H_2_ content higher than 80%vol regardless of the butiá fraction used as raw material. The biochars presented specific surface areas reaching 220.4 m^2^ g^−1^. Additionally, the endocarp-derived biochar presented a CO_2_ adsorption capacity of 66.43 mg g^−1^ at 25 °C and 1 bar, showing that this material could be an effective adsorbent to capture this greenhouse gas. Moreover, this capacity was maintained for 5 cycles. Biochars produced from butiá precursors without activation resulted in a higher surface area and better performance than some activated carbons reported in the literature. The results highlighted that pyrolysis could provide a green solution for butiá agro-industrial wastes, generating H_2_ and an adsorbent for CO_2_.

## 1. Introduction

Pollution is one of the major global issues regarding the environment. The main source of this problem is greenhouse gas emissions [1]. Industrialization and the combustion of coal and fossil fuels result in the emission of greenhouse and toxic gases [2], such as carbon dioxide (CO_2_), which causes harm to human health, and also contributes to global warming [3]. The increased CO_2_ concentrations in the atmosphere have warmed the planet substantially [4]. Recent measurements found a mean concentration of 400 ppm [5,6,7] representing an elevation of 100 ppm compared to its pre-industrial time [8]. Therefore, reducing greenhouse gas emissions is one of the greatest global challenges through 2050 [9].

There are different methods to remove carbon dioxide, including absorption [10], cryogenic separation [11], membrane [12], and adsorption [13,14,15,16,17]. Adsorption is a favorable method due to its simple operation, low energy consumption, and low equipment cost [18]. This technology has been regarded as one of the most promising for mitigating greenhouse gases [19]. Adsorbents for CO_2_ show advantages such as wider temperature range operation, less harmful disposal, yield, less waste generation, and weak bonding with CO_2_, resulting in lower regeneration energy [20]. Among different materials for CO_2_ adsorption, such as activated carbon (AC) [13,21,22,23], biomass/biowaste [24,25], zeolite [26,27], graphene [28], metal-organic frameworks [29,30], biochar [31,32] has been highlighted in recent years.

Biochar is the solid fraction obtained from biomass pyrolysis [33]. Besides biochar, pyrolysis generates non-condensable gases (CO, CO_2_, CH_4_, and H_2_) and condensable gases (bio-oil/liquid phase) [34]. Biochar presents a porous structure with abundant functional groups on the surface [35,36]. Producing biochar using waste biomass as a precursor is an effective process for converting waste into a high-value-added product [37]. The use of biomass has emerged as a promising low-cost solution [38] for the treatment and management of large volumes of agro-industrial wastes [39,40]. In this sense, biochar comes from a wide range of biomass, such as fruit, legume peels and husks [41,42,43,44,45], bagasse/pomace, fruit pit and shells [46,47,48,49,50], forestry wastes and pruning [51,52,53], sludge [54,55] and animal manure [56,57]. The biochar obtention releases more hydrogen than it consumes, making it a negative-emission technology [58]. This technology could be one cost-effective and environmentally friendly method for mitigating climate change [19] and pollution from inappropriate solid waste management.

The genus *Butia* belongs to the *Arecaceae* family, is native to South America, occur naturally in Brazil, Uruguay, Argentina, and Paraguay, and has great potential for expansion. Brazil retains the majority of species, and the occurrence spans Bahia, Santa Catarina, and Goiás, but most populations are found in the Rio Grande do Sul [59]. Butiá fruits as food are ancestral and are consumed fresh or used as culinary ingredients in juice, jelly, ice cream, yogurts, sweets, flour, and liquor [60,61,62,63]. Processing native fruits, such as butiá, is an economic activity for smallholders and farmers [60] that offers the highest potential for income generation [64]. The pulp contains high concentrations of vitamins (C and A), fibers, and phenolic compounds, potentially expanding the agro-industrial use of the fruits [61]. In addition, extracts of butiá fruits have shown potential antimicrobial properties [65,66]. Seed oil was studied to produce biofuel [67,68] and antibiofilm activity [69]. Besides the fruits and leaves used, the seeds are usually discarded [70], motivating studies to recover this waste.

Sustainable use of butiá palm groves could increase family incomes and result in social and environmental dividends [60]. However, this development would generate agro-industrial solid wastes that need appropriate treatment for energy generation or producing material for environmental regeneration. Recent studies have reported butiá endocarps [71,72]. However, no works were founded employing other fractions from butiá wastes, such as fibers and almonds, as a precursor of adsorbents, highlighting this work’s scientific contribution/novelty. Considering air and soil pollution, from the perspective of sustainable development, it is interesting to obtain solid biochar from butiá wastes that can capture CO_2_.

In this paper, butiá agro-industrial wastes were pyrolyzed, aiming to produce biochar for CO_2_ adsorption. The pomace obtained in familiar agroindustry was separated into four precursors (fibers, endocarps, almonds, and deoiled almonds). The pyrolysis was conducted at 700 °C, and the yield of solid, liquid, and gaseous fractions was evaluated. Furthermore, the precursors and the pyrolysis products were characterized. Finally, the potential of the produced biochar for CO_2_ adsorption was studied.

## 2. Results and Discussion

### 2.1. Features of the Butiá Precursors

Table 1 shows the characterization of four precursors separated from butiá pomace. The moisture content for precursors ranged from 5.19 to 6.06%wt. It was similar to other lignocellulosic biomasses such as switchgrass (6.25%wt.) [73] and palm fiber (4.23%wt.) previously dried in the thermochemical process [74]. Moreover, the results for butiá precursors were lower than tucumã seed (7.6%wt.) [75], almond and walnut shell (7.7 and 11%wt.) [76], and açaí seeds [77], which are similar biomasses. The difference observed could be explained by the dry process initially used for butiá biomass conservation. Moisture content less than 10%wt. is desirable for biomass conversion in thermochemical processes [78].

Ash content presented in the literature on another lignocellulosic material range from 0.25 to 2.82%wt. [59,74,76,79]. In this work, the samples FIB and ALM presented similar ash content to tucumã (1.45%wt.) [75], açaí seeds (1.36%wt.) [80] and almond shells (1.30%wt.) [81]. Comparing DOA with ALM, the higher content of ashes could be explained by the oil extraction from almonds, resulting in a higher presence of ashes (2.25%wt.) and a lower percentage of extractives (1.35%wt.).

For good power generation potential, the fixed carbon content should range from 15 to 25%wt. for efficient burning [82]. As can be seen in Table 1, samples FIB and END presented fixed carbon at this range and showed agreement with rubber seed shells (23.4%wt.) [83], avocado stone (22.4%wt.) [84] and Brazilian nut (25.21%wt.) [85]. Otherwise, samples ALM and DOA following rice straw (8.1%wt.) [86], sawdust (7.74%wt.) [87], and algae (12.8%wt.) [88]. Low ash content and high fixed carbon show that biomass is a potential candidate for bioenergy production [89]. Among the evaluated materials, FIB and END showed the greatest potential for this.

Related to volatile matter (Table 1), Aguiar et al. (2014) [90] reported that in butiá fruits, this parameter depends on maturation grade, climatic conditions, storage time and conditions, and specie of plant. The volatile matter results (Table 1) for the samples FIB and END corroborated tucumã (78.64%wt.) [75], banana trunk (74.33%wt.) [91], and olive kernel (75.8%wt.) [32], while DOA is near to the content of palm shells (62–85%wt.) [92], which has high oil yield like butiá almonds and palm fibers (86.51%wt.) [93] and ALM is near sawdust (90.92%wt.).

Table 1 revealed that the DOA sample resulted from a higher cellulose content. END presented higher lignin, while FIB had the major hemicellulose content. Percentages of hemicellulose and lignin obtained for FIB are comparable to wheat straw (23 to 30%wt. and 12 to 16%wt.), rice husk (12 to 29.3%wt. and 15.4 to 20%wt.), and sugarcane bagasse (12 to 29.3% and 1.4 to 20%wt.) [94]. END presented similar hemicellulose and lignin content to palm kernel shells (23.82 and 45.59%wt., respectively) [95]. ALM sample showed lignin content comparable to tucumã seeds (19.91%wt.) [77], hemicellulose content, and extractive similar to the almond hull (9.0 and 36.25%wt., respectively) [96].

Thermogravimetric curves (TG) and derivatives from the thermogravimetric curve (DTG) are shown in Figure 1a–d. The TG/DTG curves presented a consistent profile for lignocellulosic biomass. At different stages of weight loss, peaks in DTG curves can be seen that refer to the release of water or volatile organic compounds and thermo-decomposition of cellulose, hemicellulose, and lignin [77]. Weight losses observed until 184 °C can be attributed to water evaporation and degradation of organic compounds biomasses, as Baroni et al. (2015) [75]. For temperatures above 200 °C, the behavior of DTG curves is predominantly exothermic due to the decomposition of hemicellulose, cellulose, and lignin into four precursors.

Comparing the samples, FIB (Figure 1a), which has the higher hemicellulose content (Table 1), starts the thermal degradation at a lower temperature. Yang et al. (2007) [97] reported that this component starts its degradation in the range of 220 to 315 °C. Poletto et al. (2012) [98] detected hemicellulose degradation by a DTG peak near 300 °C. In this work, this peak is observed in DTG curves for the four precursors but is most pronounced in FIB (Figure 1a) and DOA (Figure 1d). END (Figure 1c) has a higher lignin content, which may explain the lower weight loss, remaining 21.3%wt. of the sample at the final temperature (800 °C). The remaining mass is similar to the açai seed (21.6%wt.), as Santos et al. (2020) [77] observed. Lignin is the major component of converting lignocellulosic wastes to char by its thermal resistance [97]. This component is responsible for delaying degradation [34], although this process starts at lower temperatures, in a wide range, from 160 to 900 °C [97]. END has pronounced hemicellulose content (Table 1) and presented (Figure 1c) degradation at 207 °C, with two peaks, at 270 and 322 °C. Perondi et al. (2017) [34] detected hemicellulose degradation from 260 to 400 °C, which is near the limit (410 °C) reported by Santos et al. (2020) [77]. 

ALM is the precursor with the highest cellulose content (Table 1) and starts degradation at 220 °C (Figure 1b), presenting peaks at 265 and 377 °C. The interval from 310 to 405 °C corresponds to 53.6% of losing weight. Subsequently, from 405 °C in a large range until 800 °C, the weight loss rate reduces drastically, which could be associated with lignin degradation (Perondi et al. 2017) [34].

The FTIR vibrational spectra of butiá precursors are reported in Appendix A. All the precursors showed bands at 3400 cm^−1^ that can be attributed to the O–H stretching vibrations of carboxylic acids, alcohols, phenols, or adsorbed water [99,100,101]. The four precursors exhibited bands at 2925 and 2854 cm^−1^ that can be assigned to C–H stretch [72,97,102]. At 1745 cm^−1^, samples ALM (Appendix A) and END (Appendix A) presented a more pronounced band, attributed to O–H stretch and C=O stretching vibration from carbonyl/carboxylic acid [102]. These samples have the highest lignin content (Table 1) and presented a more intense band at 1457 cm^−1^. This signal is attributed to the CH_2_ stretching deformation of lignin [103]. All the precursors present a band at 1160 cm^−1^, assigned to asymmetric stretching C–O–C in cellulose [103]. The band at 1040 cm^−1^ is related to the C–O stretch in carbohydrates (cellulose, hemicellulose, and lignin) [91,104]. Vibrations between 450 and 900 cm^−1^ are characteristic of aromatic ring C–C stretching [91].

XRD diffraction patterns are shown in the Appendix A, and no crystalline phase was observed in all precursors. Then, it can be stated that butiá endocarps, fiber, almonds, and deoiled almonds have an amorphous structure. The Appendix A also presents SEM micrographs of FIB, END, and DOA (Appendix A). The micrographs presented roughness at the surface for FIB and cavities for END and DOA samples. The high oil content in ALM made SEM analysis impossible for this sample. 

### 2.2. Results of the Pyrolysis Process

The product yields obtained from the butiá precursor’s pyrolysis (END, FIB, ALM, and DOA) are shown in Figure 2. The sample END presented the highest biochar yield (31.9%wt.) due to its major lignin content. Lignin fragments have multiple aromatic rings that crosspolymerize to form more carbonaceous solids [105]. According to Wan et al. (2022) [40], lignin presents lower degradation as a consequence of its thermal resistance, resulting in a higher amount of biochar [20]. Sample ALM resulted in a higher bio-oil yield (72.2%wt.). This result may be explained by the high content of oil available in butiá almonds, ranging from 30 to 57.8%wt. [69,106,107] Considering their composition (Table 1), DOA and ALM presented the highest volatile matter content. Ahmad et al. (2017) [78] reported that a high content of volatile matter is expected to favor the formation of liquid and gaseous in pyrolysis. The ALM sample showed the highest bio-oil yield and, consequently, the lowest biochar yield (12.1%wt.). Low solid yield can make it unfeasible when the pyrolysis intents to obtain adsorbents. In addition, the oil obtained from butiá almonds extracted by hexane has anti-biofouling properties against total microorganisms, aciduric bacteria, lactobacilli, and *Streptococcus mutans* [108]. Butiá almond oil extracted by hexane has also been studied to produce biofuels [67,68]. DOA pyrolysis yielded a better biochar yield (19.8%wt.) than ALM. 

The increase in the cellulose and hemicellulose content leads to higher production of volatile vapor in pyrolysis [20]. This observation explains why DOA (69.6%wt. of cellulose and hemicellulose) had the highest gas production, reaching 50.5%wt. In addition, in pyrolysis, cellulose produces more fragments of smaller molecular sizes, increasing liquid and gaseous fractions [105].

### 2.3. Non-Condensable Gases Generation 

Regarding gas production, Figure 3 presents the distribution of non-condensable gases affected by temperature. For all precursors, the major gas produced is H_2_ which tends to increase in generation with temperature. This tendency was also observed by Perondi et al. (2017) and De Conto et al. (2016) [34,109]. Nevertheless, compared to other samples employed in this work, only END (Figure 3c) still increased H_2_ generation 30 min after reaching the isotherm temperature (700 °C). This trend is attributed to lignin decomposition [97]. The H_2_ generation should be highlighted since it is a green gas and could be used as a fuel.

CO generation was favored at lower temperatures until 600 °C, highlighted by sample END, and can be assigned to the thermal decomposition of hemicellulose [97]. At 700 °C, CO generation is observed for FIB (Figure 3a), ALM (Figure 3b) and END (Figure 3c). This observation can be related to the second pyrolysis of solid material and lignin degradation [97]. 

CH_4_ generation was observed mainly until 600 °C, by degradation of the methoxyl group, decreasing after 700 °C. END sample (Figure 3c) released CH_4_ at 700 °C (at 0 and 30 min) to the high content of lignin [34]. This tendency was also confirmed by De Conto et al. (2016) [109].

For CO_2_ production, it was observed that it decreases with temperature, following Chang et al. (2016) [110]. Yang et al. (2007) [97] demonstrate that 500 °C results in higher CO_2_ production. However, CO_2_ releases still at 600 °C and 700 °C (Figure 3c), which can be attributed to hemicellulose, but majorly, lignin degradation [97].

### 2.4. Biochars Characteristics

Appendix A shows FTIR results obtained from biochars. For all biochars, the bands’ intensity indicated a strong reduction. Similar behavior was observed previously [99,100]. For example, in Appendix A, it can be seen that a band as 3445 cm^−1^ signed to O–H stretching vibration remained [97]. Additionally, some bands around 1700 cm^−1^ remained. In general, the amount and intensity of the bands decreased after pyrolysis operation regardless of precursors. This trend results from volatilizing a great fraction of the precursors during the thermochemical process. The volatilization of these groups causes an improvement in the textural features of the biochars compared to their precursors, being favorable for adsorption purposes.

XRD analyzed the biochars of butiá precursors, and the diffraction patterns are shown in the Appendix A. No crystalline phase was observed for all biochars, indicating an amorphous structure. This non-regular structure is generally adequate for adsorption purposes [111] since it provides more empty spaces and allows the accommodation of molecules at the solid surface.

Figure 4 shows the micrographs of biochars. These micrographs show cavities that can indicate a thermal degradation reaction during the pyrolysis process, releasing volatile matter and resulting in a visible porous surface, which was later confirmed through pore distribution analysis. In addition, these SEM micrographs confirm that pyrolysis provides modifications into butiá precursors and shows bumps, cavities, and grooves that are favorable for adsorption purposes [99].

N_2_ sorption/desorption analysis was performed, and the isotherms are presented in Appendix A. The BET and BJH results are presented in Table 2. From Appendix A, it can be seen that samples FIB.700 (a) and DOA.700 (d) presented type I isotherm, characteristic of solids with microporosity [112]. Nonetheless, Table 2 shows that these biochars have an average pore size slightly higher than 2 nm, the limit for micropores [113]. This range of pore diameter is called narrow mesopores (<2.5 nm) [114] and also presents type I isotherms. Pore size distribution (Appendix A) confirms that FIB.700 (a) and DOA.700 (d) presented narrow mesopores. FIB.700 and END.700 (Appendix A presented non-reversible isotherms attributed to these materials’ intrinsic and typical characteristics. Isotherms shapes in Appendix A are usual for chars and may be explained by the presence of necks. Better results obtained for biochars from FIB and DOA may be explained by the lowest lignin content, already presented in Table 1. Lignin may inhibit porosity [115], resulting in dense solids. These results are similar to those obtained by El-Gamal et al. (2017) [46] for biochars obtained from sugarcane bagasse (0.11 cm^3^ g^−1^ and 2.31 nm). Additionally, higher cellulose contents may result in a microporous structure formation [51,116,117]. This statement explains the higher pore volume presented by DOA.700, which has more than 50% of cellulose content. The isotherms observed for ALM.700 and END.700 are type IV and II, respectively. These isotherm shapes are characteristic of non-porous solids or relatively higher pores [112]. END.700 and ALM.700 presented the larger pores and the lowest total pore volumes according to isotherm types (Table 2).

The highest values for the specific surface area were from FIB.700 and DOA.700 biochars, reaching 183.59 and 220.43 m^2^ g^−1^, respectively. These surface area values are higher than chars modified or activated chars from date stone (187 m^2^ g^−1^) [118], beer solid wastes (80.5 m^2^ g^−1^) [100], corn stover (24–129 m^2^ g^−1^) [119], and freshwater sludge (96–285.78 m^2^ g^−1^) [120]. Sample ALM.700 (Table 2) resulted in a lower surface area value. However, the literature reports similar values for other biochars obtained from different materials such as sweet lime (1.9 m^2^ g^−1^) [121], oak bark (1.9 m^2^ g^−1^) [122], and chicken manure wastes (0.98–4.9 m^2^ g^−1^) [56]. Li et al. (2005) [123] presented that specific surface areas of your biochars ranged from 65 to 95 m^2^ g^−1^. Considering this information, END.700 is also in the literature range. Comparing results obtained for surface area for ALM.700 and DOA.700 (Table 2), the previous oil extraction allows the use for other purposes, which increases the waste’s added value and results in better properties for the biochar to act as an adsorbent.

### 2.5. Results of CO_2_ Capture on the Biochars

The CO_2_ adsorption capacity of biochars was tested at 25 °C and atmospheric pressure. The results are shown in Figure 5. END.700 presented a higher adsorption capacity (66.43 mg g^−1^), and the total adsorption capacity was reached practically in the first minute of the run (58.65 mg g^−1^) with an increase of 11.7%, from 1 to 11.8 min. Botomé et al. (2017) [124] obtained similar capacity and behavior by employing activated carbon from CCA-treated wood to adsorb CO_2_ at the same pressure and temperature. It is known that a high surface area and pore volume are needed to improve CO_2_ adsorption [125], and this process is dominated by micropores [31]. In parallel, at pyrolysis, the higher lignin content inhibits pore formation [115], and the END precursor has the higher lignin content (Table 1). Consequently, END.700 biochar has a low micropores volume (Table 2). Although micropore volume and surface area are important to CO_2_ adsorption, alkalinity strongly influences this process [31]. Table 2 presents the results for pH_PCZ_. END.700 shows the highest value (7.34) at the point of zero charges [126]. This observation may explain why END.700 presented a higher capture than other biochars. CO_2_ is a weak Lewis acid gas, which can interact with the alkaline adsorbent, and lignin is a Lewis base. Thus, the higher content of lignin may explain the increase in CO_2_ capture. The pH_PCZ_ seems to play a secondary role in CO_2_ adsorption, Wjihi et al. (2021) [127] reported that a higher pH_PCZ_ resulted in higher adsorption capacity. FIB.700 showed an adsorption capacity of 54.59 mg g^−1^, ALM.700 presented 48.87 mg g^−1,^ and DOA.700 exhibited an adsorption capacity of 51.76 mg g^−1^, respectively (Figure 6).

Considering that surface area impacts CO_2_ adsorption [128], this result can be evaluated in terms of CO_2_ mass per unit of area of adsorbent (mg CO_2_ m^−2^) [124] bo. The best performance (per unit of area) was reached by ALM.700, which presented an adsorption capacity of 25.45 mg CO_2_ m^−2^, higher than marine shale with 14.948 mg CO_2_ m^−2^ [129]. Appendix A presents a compilation datum of adsorbents from literature to compare with biochars obtained in this work. As can be seen, the biochars produced from butiá precursors pyrolysis without the activation step resulted in a better performance than some activated carbons with large surface area and activation.

Figure 6 shows the cycles of CO_2_ adsorption/desorption for biochars obtained from butiá precursors. The biochars presented fast adsorption and desorption, which is an important feature in selecting an adsorbent. Biochars presented suitable stability for adsorption and desorption cycles, evaluated by the same behavior in the 5 cycles, similar to those presented by Botomé et al. (2017) and Singh et al. (2019) [124,130]. At the first peak, END.700 (Figure 6c) presented 66.43 mg g^−1^. This biochar presented an average capacity of 65.41 mg g^−1^, considering the five peaks, representing a regeneration of 98.5%. The five peaks of FIB.700 (Figure 6a), ALM.700 (Figure 6b) and DOA.700 (Figure 6d) exhibited average capacity of 54.86 ± 0.63 mg g^−1^, 49.91 ± 0.78 mg g^−1^ and 52.83 ± 1.02 mg g^−1^, respectively, and regeneration of 100%. Total regeneration for CO_2_ adsorbents is also reported by Botomé et al. (2017) and Li and Xiao (2019) [124,131].

## 3. Material and Methods

### 3.1. Obtainment and Pre-Treatment of the Precursors

The butiá wastes used in this work were donated by familiar agroindustry (Sete de Setembro city, Rio Grande do Sul, Brazil (−28°12′30.7”, −54°29′31.7”)). The pomace was collected after a pulp extractor and was dried in an oven (SP-100/216, SPlabor, Brazil) at 60 °C for 48 h. The dried biomass was separated into fibers and seeds. The seeds were broken to remove the almonds. Fibers (FIB), endocarps (END), and almonds (ALM) were fragmented in a knife mill (DeLeo 0416, Porto Alegre, Brazil) for particle sizes less than 2 mm. Some almonds were separated to extract lipids by Soxhlet in two steps, according to TAPPI, 1997 [132]: firstly, using hexane (Cinética, Brazil) (30 g of almonds to 300 mL of solvent) for 5 h. After, the solids were dried in an oven for solvent removal. The second extraction step was conducted with a mixture of ethanol (Audaz, Brazil) and benzene (Cinética, Brazil) (1:2, *v*/*v*) for 5 h. Finally, solvents were recovered using a rotary evaporator. The solids obtained after extractions correspond to a deoiled almond (DOA) precursor.

### 3.2. Characterization of the Precursors

The four precursors (FIB, END, ALM, and DOA) were characterized using proximate analysis based on the D3172-89 (1993) standard from the American Society for Testing and Materials [133]. Van Soest’s gravimetric method determined the cellulose, hemicellulose, and lignin composition [134]. Fourier transform infrared spectroscopy analysis (Shimadzu, Prestige 21, Japan) was performed to identify functional groups from 4500 to 400 cm^−1^. X-ray diffraction (Rigaku, Miniflex 300, Shibuya, Japan) was used to analyze the crystalline or amorphous nature of the samples, applying a Cu Kα radiation (λ = 154,051 Å) at 20 mA and 30 kV by scanning from 5 to 70°, with a step of 0.03°. The thermal stability of the samples was analyzed by Thermogravimetrical analysis (STA 449 F3, Jupiter, Netzsch, Selb, Germany) with 10 mg of sample, a heating rate of 5 °C min^−1^ from ambient temperature to 800 °C and nitrogen flow (50 mL min^−1^). The morphology was analyzed by Scanning electron microscopy (Tescan Mira 3, Kohoutovice, Czech Republic), using 5 kV.

### 3.3. Pyrolysis Process

The pyrolysis of FIB, END, ALM, and DOA was performed in a tubular reactor described and reported by Perondi et al. (2017) [34]. The experiments were conducted at a 5 °C min^−1^ heating rate until 700 °C, at 200 mL min^−1^ N_2_ flow, for 100 g of precursor (FIB, END, ALM, or DOA). A 60 min bolding time was used after the final temperature (700 °C) was reached. The gas samples were collected in a non-isothermal region (500, 600, and 700 °C points) and an isothermal region (after 30 and 60 min after 700 °C was reached). The pyrolysis parameters were selected considering previous studies [124] for better biochar and H_2_ generation. The gases and vapors sampling was conducted, as reported by Perondi et al. (2017) [34]. A gas meter was used to measure the volume of gas produced. The bio-oil at boilers and the biochar (residual solid) were collected, and their masses were determined for yield computation. The percentage yields of biochar (*R_biochar_*), pyrolytic liquid (*R_liquid_*), and non-condensed gases (*R_gas_*) were determined as follows:(1)$$R_{biochar}\left(\%\right)=\frac{m_{biochar}}{m}\times 100$$
(2)$$R_{liquid}\;\left(\%\right)=\frac{m_{liquid}}{m}\times 100$$
(3)$$R_{gas}\;\left(\%\right)=100-R_{liquid}-R_{biochar}\;$$
where *m* (g) is the mass of precursor material inserted into the reactor, *m_biochar_* (g) is the residual solids mass in the reactor after the pyrolysis process, and *m_liquid_* (g) is the total mass of liquid in the collectors after pyrolysis process.

### 3.4. Non-Condensable Gases Characterization

The non-condensable gases (H_2_/CO/CO_2_/CH_4_) analysis was performed by a gas chromatograph (Dani Master GC) equipped with a thermal conductivity detector. A capillary column Carboxen^TM^ model 1006 (SUPELCO), with a length of 30 m, 0.53 mm internal diameter, and 30 mm film thickness, was used. The calibration curves were constructed from gas standards.

### 3.5. Biochars Characterization

Solid fractions obtained by pyrolysis of FIB, ALM, END, and DOA were named FIB.700, ALM.700, END.700, and DOA.700, respectively. These biochars were characterized by FTIR, XRD, and SEM, using the same procedure reported in Section 2.2. The N_2_ adsorption–desorption isotherms at −196 °C were obtained in a surface area and porosimetry analyzer (Quantachrome Instruments, Nova 1200) by degassing biochars for 20 h under a vacuum. The surface area was determined by Brunauer Emmett-Teller (BET). The total pore volume was determined from the amount of nitrogen adsorbed at P/P0 = 0.99. The micropore area and the micropore volume were estimated by t-plot. N_2_ isotherms and non-local density functional theory (NLDFT) determined the pore size distribution. The point of zero charges (pH_PZC_) was determined by the eleven points experiment as follows: Eleven flasks with 50 mL of a solution containing 50 mg of the samples (initial pH values in the range from 1.0 to 11.0, which were adjusted with HCl and NaOH) were stirred at 120 rpm (Fanem, 315 SE, São Paulo, Brazil) for 24 h. The pH values were measured before and after the agitation (Marte, MB10, São Paulo, Brazil).

### 3.6. Biochars Potential for CO_2_ Capture

The biochars FIB.700, END.700, ALM.700, and DOA.700 were used in adsorption to capture CO_2_. CO_2_ adsorption studies were carried out in thermogravimetric equipment (Netzsch, STA 449 F3, Jupiter, Selb, Germany), employing 10 mg of biochar, with the same methodology reported by Botomé et al. (2017) [124]. When the point of adsorbent saturation was achieved, CO_2_ flow was interrupted, and the sample was heated until 120 °C, under N_2_ flow, to desorption. The procedure was repeated until complete five cycles of adsorption–desorption.

## 4. Conclusions

A possible route to valorize butiá wastes was studied in this work. The butiá wastes were divided into fibers, endocarps, almonds, and deoiled almonds, and these fractions were pyrolyzed. The pyrolysis yields varied according to the precursor type. The high yields in terms of biochar (31.9%wt.), bio-oil (72.2%wt.), and gases (50.5%wt.) were found, respectively, for endocarp, almonds and deoiled almonds as precursors. The main gas released was H_2,_ regardless of the precursor type. The generated biochars presented interesting features to CO_2_ capture (66.43 mg g^−1^ or 25.45 mg CO_2_ m^−2^), and these characteristics were maintained for 5 cycles. From the cleaner production perspective, the pyrolysis of butiá agro-industrial wastes presented interesting possibilities: power generation by gases and oil released at the process; aggregate value to precursors through the biochar preparation, contributing to solid wastes management, and CO_2_ capture by the produced biochar.

## Figures and Tables

**Figure 1 molecules-27-07515-f001:**
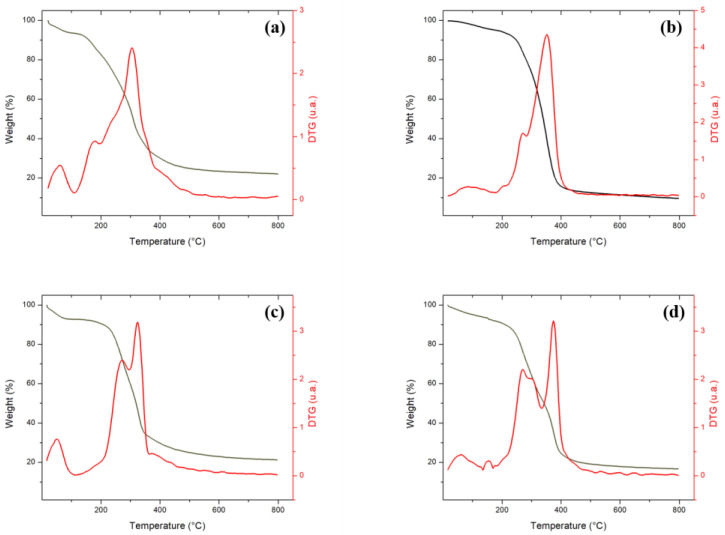
Thermogravimetric and derived curves to precursors (**a**) FIB, (**b**) ALM, (**c**) END, and (**d**) DOA.

**Figure 2 molecules-27-07515-f002:**
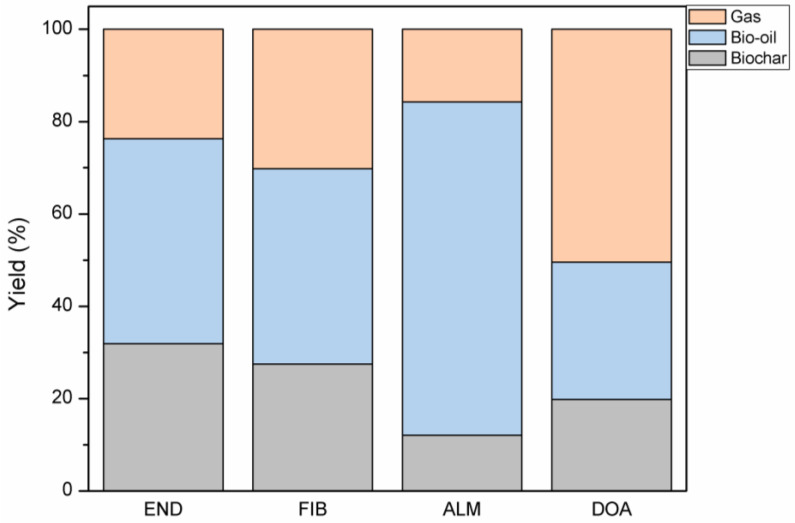
Product yields from pyrolysis of END, FIB, ALM, and DOA at 700 °C.

**Figure 3 molecules-27-07515-f003:**
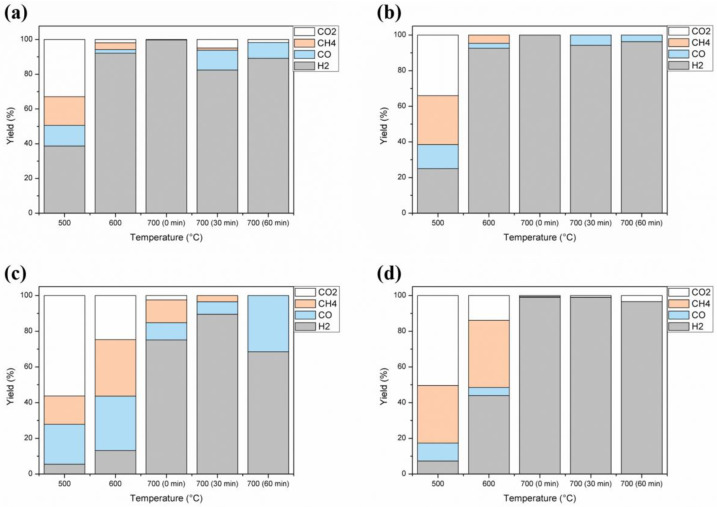
Effect of temperature on gas production during the pyrolysis of (**a**) FIB, (**b**) ALM, (**c**) END, and (**d**) DOA.

**Figure 4 molecules-27-07515-f004:**
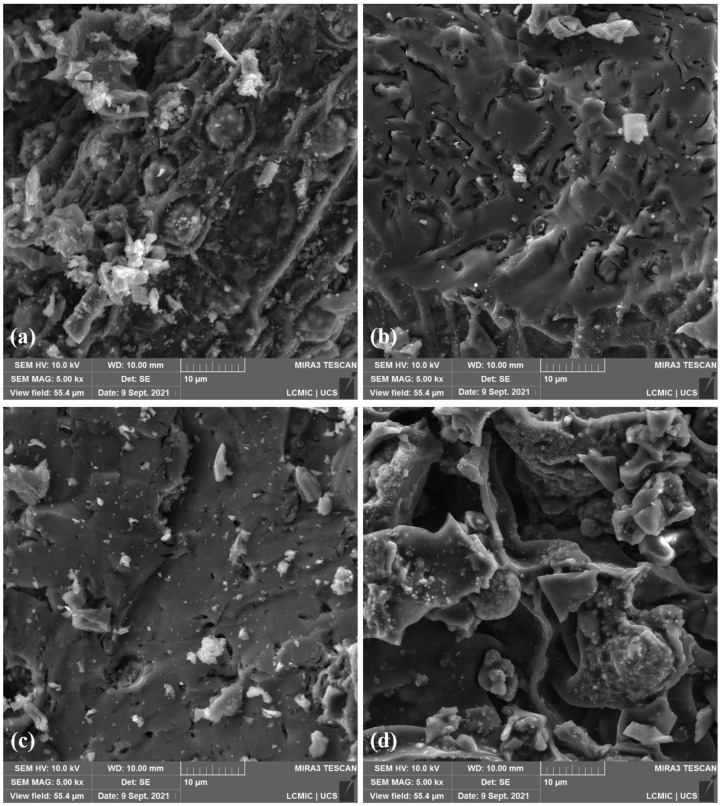
SEM micrographs of (**a**) FIB.700, (**b**) ALM.700, (**c**) END.700 and (**d**) DOA.700.

**Figure 5 molecules-27-07515-f005:**
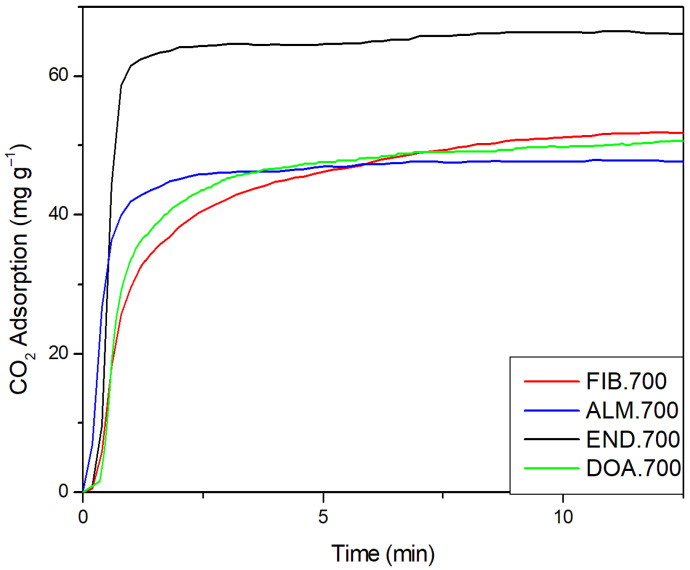
CO_2_ adsorption capacity of biochars from FIB, ALM, END, and DOA.

**Figure 6 molecules-27-07515-f006:**
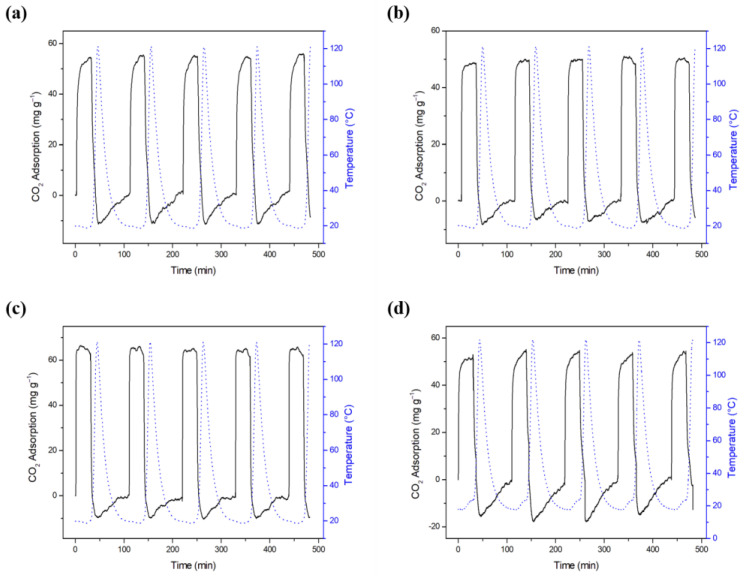
CO_2_ adsorption cycles in biochars (**a**) FIB.700, (**b**) ALM.700, (**c**) END.700 and (**d**) DOA.700.

**Table 1 molecules-27-07515-t001:** Proximate analysis results and cellulose, hemicellulose, and lignin composition for the butiá precursors.

Proximate Analysis *				
	FIB	END	ALM	DOA
Moisture (%wt.)	5.48 ± 0.08	5.19 ± 0.20	5.22 ± 1.44	6.06 ± 0.19
Ash (%wt.)	1.72 ± 0.08	0.51 ± 0.02	1.15 ± 0.06	2.25 ± 0.14
Fixed carbon (%wt.)	22.25 ± 0.58	25.6 ± 0.41	7.62 ± 0.42	12.13 ± 0.06
Volatile matter (%wt.)	76.03 ± 0.51	74.06 ± 0.27	91.23 ± 0.41	85.62 ± 0.18
**Chemical Composition ***				
	**FIB**	**END**	**ALM**	**DOA**
Cellulose (%wt.)	5.94 ± 0.68	6.14 ± 1.35	10.57 ± 0.38	57.79 ± 0.39
Hemicellulose (%wt.)	29.00 ± 1.14	24.34 ± 1.17	8.03 ± 0.36	11.81 ± 0.44
Lignin (%wt.)	15.32 ± 0.34	48.14 ± 1.44	19.12 ± 1.14	10.80 ± 0.85
Extractive (%wt.)	11.10 ± 0.65	7.44 ± 0.38	36.62 ± 1.14	1.35 ± 0.35

* All results are mean ± standard error for *n* = 3.

**Table 2 molecules-27-07515-t002:** Textural characteristics of the butiá-based biochars.

Sample	Surface Area BET (m^2^ g^−1^)	Average Pore Size (nm)	Micropores Volume (cm^3^ g^−1^)	Mesopores Volume (cm^3^ g^−1^)	Total Volume Pores (cm^3^ g^−1^)	pH_PCZ_
FIB.700	183.59	2.527	0.109	0.0	0.109	6.52
ALM.700	1.92	5.628	0.0	0.00194	0.00194	7.20
END.700	58.39	3.336	0.065	0.0	0.065	7.34
DOA.700	220.43	2.456	0.123	0.0	0.123	7.05

## Data Availability

Data available upon reasonable request.

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
