# Peer review of "Valorization of Different Fractions from Butiá Pomace by Pyrolysis: H2 Generation and Use of the Biochars for CO2 Capture"

_molecules, 2022, doi:10.3390/molecules27217515_

Round 1

Reviewer 1 Report

The manuscript can be accepted after the author provides additional data which can support your claim. The following points need to be addressed before the manuscript can be accepted:

1) SEM and its scale bar are not clear.

2) Provide Nitrogen adsorption/desorption plot which you use to calculate surface area. Surface area is not good to call this material as a good adsorbent for CO2 capture.

3) More characterization such as TEM, DSC etc. can provide more information about the size of particles and thermal stability.

Author Response

The manuscript can be accepted after the author provides additional data which can support your claim.

Dear reviewer, the authors would like to thank you for the review. We hope that all questions related have been cleared and that the manuscript has been improved for approval.

The following points need to be addressed before the manuscript can be accepted:

1) SEM and its scale bar are not clear.

Response: Dear reviewer, Figure 4 was updated in order to clarify the details and scale of SEM micrographs. Thank you for your suggestion.

2) Provide Nitrogen adsorption/desorption plot which you use to calculate surface area. Surface area is not good to call this material as a good adsorbent for CO2 capture.

Response: Dear reviewer, the N2 sorption/desorption isotherms used to calculate surface area are presented in Figure 6S (supplementary material). About relation between surface area and CO2 capture, although surface area is important to CO2 adsorption, superficial characteristics of adsorbent influences this process. This observation may explain why the biochar obtained from endocarps (END.700 - 58.39 m² g-1) presented a higher capture than other biochars. FIB.700 (183.59 m² g-1) shown an adsorption capacity of 54.59 mg g-1, ALM.700 (1.92 m² g-1) presented 48.87 mg g-1, and DOA.700 (220.43 m² g-1) exhibited an adsorption capacity of 51.76 mg g-1, respectively (Figure 6).

3) More characterization such as TEM, DSC etc. can provide more information about the size of particles and thermal stability.

Response: Dear reviewer, first of all we would like to thank you for your suggestion. Due to the challenges empoused by covid-19, these analyses couldn’t be performed. At this moment, the equipment is out of operation, which makes attempts impossible.

Related to the evaluation of relevance of the references cited in introduction, we would like to announce that we reviewed the number of references, leaving behind 11 works that were slightly relevant to our research. Remain in the text the essential studies to understanding the background of this study.

Reviewer 2 Report

Regarding the manuscript entitled “Valorization of different fractions from Butiá pomace by pyrol- 2 ysis: H2 generation and use of the biochars for CO2 capture” in which the authors investigated the pyrolysis of fractions of the pomace, like fibers, endocarps, almonds, and deoiled almonds for producing biochar for CO2 adsorption. this research has high environmental impact for CO2 remediation. The prepared biochar is characterized and exhibit proper porosity and structure which facilitate the CO2 adsorption/desorption process. The whole manuscript is well presented. I just have few comments:

-          Why author select 700ËšC for pyrolysis?

-          For Figure 1S, add the detected groups to the figure.

-          Why the author did not apply the biochar for adsorption of various pollutants such as Sox or NOx?  

Author Response

Regarding the manuscript entitled “Valorization of different fractions from Butiá pomace by pyrolysis: H2 generation and use of the biochars for CO2 capture” in which the authors investigated the pyrolysis of fractions of the pomace, like fibers, endocarps, almonds, and deoiled almonds for producing biochar for CO2 adsorption. This research has high environmental impact for CO2 remediation. The prepared biochar is characterized and exhibit proper porosity and structure which facilitate the CO2 adsorption/desorption process. The whole manuscript is well presented.

Dear reviewer, the authors would like to thank you for the review. We hope that all questions related have been cleared and that the manuscript has been improved for approval.

I just have few comments:

-          Why author select 700ËšC for pyrolysis?

Response: Previous studies from the same research group has shown that 700ºC provide properly production of biochar an H2 generation at pyrolysis. Botomé et al. (2017) has added as a reference at methodology (2.3 Pyrolysis Process).

M.L. Botomé, P. Poletto, J. Junges, D. Perondi, A. Dettmer, M. Godinho, Preparation and characterization of a metal-rich activated carbon from CCA-treated wood for CO2 capture, Chem. Eng. J. 321 (2017) 614–621. https://doi.org/10.1016/j.cej.2017.04.004.

-          For Figure 1S, add the detected groups to the figure.

Response: Dear reviewer, thank you for the appointment. The detected groups were added to the Figure 1S.

-          Why the author did not apply the biochar for adsorption of various pollutants such as Sox or NOx?  

Response: Dear reviewer, the authors understand and appreciate your comment. We know that SOx and NOx emissions contributes to acidification of soil and waterways, acid rain and photochemical smog, for example. Studies to capture this gases and evaluation of the synergetic adsorption of this pollutants need to be performed. However, this evaluation is not the focus of the work, and your contributions will be addressed in a future paper. At this moment, authors don’t have SOx and NOx to evaluate this performance.

Author Response

Excellent work, it fully meets all the magazine's criteria,

- Correctness of assumptions

- Appropriateness of the experiments but also analysis of the results

- Detailed description of the experimental procedures.

Dear reviewer, the authors would like to thank you for the review. We hope that all questions related have been cleared and that the manuscript has been improved for approval.

You should pay attention to the following points

-The second extraction step was conducted with a 100 mixture of ethanol (Audaz, Brazil) and benzene (Cinética, Brazil) (1:2, v/v) for 5 hours. 101 This Solvent is hazardous to human health, perhaps you should have explained better how it is recovered and if it the solvents can be replaced with more environmentally friendly solvents?

Response: We would like to thank you for your observation. The extraction of oil from almond samples was performed according to T204 cm-97 Standard (TAPPI, 1997). The use of benzene for extraction followed the procedure adopted at this standard. The recuperation of this solvent was carried out with a rotary evaporator. However, we understand that we reported that the oil obtained from butiá almonds has interesting properties against microorganisms and the use of benzene may compromise this. Nonetheless, for this applications related by Peralta et al., 2013 [69] the oil was extracted with hexane. Hexane is an organic solvent commonly used in the food industry to remove oil from food sources and also was employed to extract butiá oil to produce biodiesel, as reported by Vieira et al. (2016) [67].

In this context, additions was made to the text and the following reference were added:

TAPPI. T 204 cm-97 - Solvent extractives of wood and pulp. Technical Association of the Pulpand Paper Industry, 1997.

-The procedure was repeated until complete five cycles of adsorption-desorption. Why is the specific repeatability of 5 cycles of any particular importance? If, for example, it was 3 or 7, would it have any effect on your results?

Response: Thank you for the remark. In another study made by this same research group, Botomé et al. (2017) also repeated 5 cycles of CO2 adsorption/desorption. Previous studies were conducted with 10 adsorption/desorption cycles, and the recovery was up to 98%. Considering the energy and gases costs, we adopted 5 cycles repeatability as a standard for this analysis. At this current study, 5 cycles presented minimum recovery 98.5% of the adsorption capacity for endocarps and 100% for fibers, almonds and deoiled almonds.

-The high oil content in ALM made SEM analysis impossible for this sample. Yes, it is understandable that there is a problem with the oil in this particular microscope, but there are probably ways to see the specific structures by removing or possibly drying the specific samples. Another alternative is to conduct the specific measurements in low temperature in order to freeze the sample and adopt a solid-like texture that would be able to be studied.

Response: Dear reviewer, we employed dried samples to perform SEM analysis and it wasn’t possible. Fragmented butiá almonds have an oil saturated surface thus, the freezing technique could be a possibility to analyze this sample. We would like to thank you for the suggestion and your contribution will be addressed in a future paper.

Based on the other reviewer comments, and considering that you suggested that results presentations must be improved, we would like to announce that we made modifications.

  • Figure 4 was updated in order to clarify the details and scale of SEM micrographs.
  • The detected groups were added to the Figure 1S at FTIR vibration spectra.

Round 2

Reviewer 1 Report

Accept in current form

Author Response

Thank you

Reviewer 2 Report

Thanks for responding to comments. 

Author Response

Thank you

Reviewer 3 Report

All necessary corrections have beeb made, I accept the publication of this article

Author Response

Thank you